# SVNC-Net: An optimized U-Net variant with 2D convolutions for lightweight 3D spleen segmentation

Mehmet Zahid Genc [1], Yaser Dalveren [2,3]*, Ali Kara [1], Mohammad Derawi[3,4], Jan Kubicek [3], Marek Penhaker[3]

**1** Department of Electrical and Electronics Engineering, Gazi University, Ankara, Turkey, **2** Department of Electrical and Electronics Engineering, Izmir Bakircay University, Izmir, Turkey, **3** Department of Cybernetics and Biomedical Engineering, VSB - Technical University of Ostrava, Ostrava, Czechia, **4** Department of Electronic Systems, Norwegian University of Science and Technology, Gjovik, Norway

* yaser.dalveren@bakircay.edu.tr

## Abstract

Accurate measurement of spleen volume is essential for the diagnosis of splenomegaly. While Computed Tomography (CT) is among the most reliable imaging modalities for this task, manual segmentation of the spleen is labor-intensive and impractical for routine clinical workflows. Automatic segmentation methods provide a more viable alternative for clinical deployment. In recent years, 3D Convolutional Neural Network (CNN) models have been widely used for this purpose due to their high segmentation accuracy. However, their computational and memory demands make them less suitable for real-time applications on edge devices with limited processing capabilities. To address these limitations, we introduce SVNC-Net (Spleen Volume and Neighborhood Convolutional Network) for efficient 3D spleen segmentation from CT scans. Rather than developing an entirely new architecture from scratch, SVNC-Net builds upon the U-Net framework with targeted architectural optimizations for efficiency. In SVNC-Net, each CT slice is processed independently using 2D convolutions. In its architecture, depthwise separable convolution is used to significantly reduce computational complexity and memory usage. To evaluate its performance and efficiency, a comparative analysis was conducted against well-known CNN-based models, including UPerNet, EMANet, CCNet, SegNet, and ShuffleNet. This evaluation was performed on two publicly available datasets used together for the first time in the literature. The promising results achieved from the comparative analysis verified that SVNC-Net is highly suitable for real-time applications and resource-constrained environments. Additionally, we explore post-training compression techniques such as pruning and quantization, which further enhance the model's compactness and inference speed. These findings contribute to the ongoing efforts to develop efficient 2D deep learning models for 3D organ segmentation, particularly in resource-constrained clinical scenarios.

**Data availability statement:** The spleen dataset used in our study is part of the Medical Segmentation Decathlon (MSD), which is widely known and frequently used in the research community. The official access point for the dataset is provided at: Project page: http://medicaldecathlon.com/ From this page, under the "Data" section, the dataset can be directly downloaded via the following link: https://goo.gl/QzVZcm, where the file Task09_Spleen.tar corresponds to the spleen dataset.

**Funding:** This study was supported by the VSB – Technical University of Ostrava in the form of a grant awarded to J.K., M.P., M.D., and Y.D. (European Union, LERCO project No. CZ.10.03.01/00/22_003/0000003, via the Operational Programme Just Transition; and project SP2025/032 'Biomedical Engineering Systems XXI') and by the VSB – Technical University of Ostrava in the form of institutional salaries for J.K., M.P., M.D., and Y.D. The specific roles of these authors are articulated in the 'Author Contributions' section. The funders had no role in study design, data collection and analysis, decision to publish, or preparation of the manuscript.

**Competing interests:** The authors have declared that no competing interests exist.

## Introduction

The spleen is one of the important organs that plays a significant role in the immune system response, as it removes waste products by filtering blood and produces white blood cells [1]. The abnormal enlargement of the spleen, known as splenomegaly, is commonly associated with cancers, infections, and other pathological conditions [2,3]. This indicates that spleen volume is the key biomarker to detect splenomegaly. Therefore, accurate measurement of spleen volume is essential for proper diagnosis [4].

In clinical applications, ultrasound (US) imaging is preferred to be used to detect splenomegaly [5]. As is known, using US yields the spleen length only. Thus, spleen volume has still been considered as the gold standard in determining the severity of diseases [6]. One of the imaging modalities that can be used to measure the volume of spleen is Magnetic Resonance Imaging (MRI) [7]. Although it is a valuable diagnostic tool, some challenges, including longer time, higher cost, or limited availability of MRI machines, need to be addressed. An alternative imaging modality is Computed Tomography (CT), regarded as the most reliable way that provides accurate measurements of spleen volume. Traditionally, the changes in the spleen volume from CT scans can be observed by an expert using manual segmentation [8]. However, the manual segmentation of the spleen is time-consuming. Instead, automatic segmentation methods could be more practical from a clinical implementation perspective. Yet, the accuracy of the first generation of the automatic segmentation methods, such as multiatlas [9] or active contours [10], is limited because of the variability in the shape of the spleen [11].

Recently, Deep Learning (DL) approaches have been used for biomedical image segmentation tasks [12]. Particularly, Convolutional Neural Network (CNN)-based models have become very popular for automatic spleen segmentation from CT scans [13–25]. It is important to note that these models are based on 3D architectures since a CT scan is inherently 3D. Even though these architectures provide high segmentation performances, their computational costs and memory needs could still be significant concerns for their usage in clinical conditions due to the limited computational capabilities of the edge devices. To address this issue, some works utilizing 2D convolutional architectures have been proposed [26–28]. Nevertheless, these works have proposed specific approaches or pipelines that can be incorporated into routine clinical workflows instead of providing a simple and lightweight 2D model for 3D spleen segmentation. This gap highlights the need for a task-specific, lightweight 2D convolutional architecture that is both computationally efficient and accurate enough for 3D spleen segmentation under standard clinical conditions. Moreover, post-training model compression techniques such as pruning and quantization have emerged as practical tools to further reduce model complexity and adapt existing architectures to embedded or real-time clinical settings.

In this study, we aim to develop a lightweight yet efficient approach for 3D spleen segmentation from CT scans. To this end, the Spleen Volume and Neighborhood Convolutional Network (SVNC-Net), an optimized and task-specific 2D variant of the

traditional U-Net architecture is proposed. The main idea behind the proposed model is to process each slice of 3D CT scans independently using 2D convolutions. A key advantage of SVNC-Net lies in its use of depthwise separable convolutions, which enhance computational efficiency by requiring fewer parameters than traditional convolutions and by reducing dependency on the number of channels. To comprehensively evaluate the segmentation accuracy and computational performance of SVNC-Net, comparative experiments are conducted against well-known CNN-based segmentation models, such as Unified Perceptual Parsing for Scene Understanding (UPerNet) [29], Expectation-Maximizing Attention Network (EMANet) [30], Criss-Cross Network (CCNet) [31], Semantic Segmentation Network (SegNet) [32], and ShuffleNet [33], on the publicly available datasets, including the Medical Segmentation Decathlon (MSD) [34] and Duke Spleen Data Set (DSDS) [35]. The results indicate that SVNC-Net achieves higher segmentation accuracy while significantly reducing training time and computational overhead. Post training compression techniques, including model aware pruning and 8 bit quantization, are applied to SVNC Net and the baseline models in addition to architectural design, and their effects on performance, memory usage, and inference time are reported. These compression techniques are applied only to the models trained on the MSD dataset, in order to analyze their computational impact without introducing dataset-specific confounders. Thus, the results achieved from this study highlight the potential of lightweight models for real-time medical image analysis and mobile diagnostic applications. The key contributions of this study can be summarized as follows:

- From a standard clinical workflow perspective, a lightweight 2D model (SVNC-Net) is proposed for 3D spleen segmentation from CT scans.

- The performances of well-known CNN-based models are comparatively assessed on the publicly available datasets for spleen segmentation.

- Useful insights that may enable the research community to develop future 2D DL models for 3D organ segmentation are provided.

- Post-training model-aware pruning and quantization techniques are applied to demonstrate further compression and speed-up without significant performance loss.

The rest of the article is structured as follows. In the following section, the relevant works proposed in the literature are discussed. The architecture of SVNC-Net section presents the network design in detail, while the Experimental details section outlines the implementation and evaluation procedures. The results achieved from the experiments are described in Results section. Next, Discussion section summarizes the key findings. Finally, the article concludes with Conclusion section.

## Related work

The implementation of 2D CNN architectures for spleen segmentation from 3D CT scans has a significant potential to enhance decision-making processes within clinical practice. By utilizing 2D CNNs, segmentation tasks can benefit from reduced computational complexity and increased efficiency compared to fully 3D approaches. However, research in this domain remains in its early stages, with only a limited number of studies addressing the development of robust and generalizable segmentation models.

An automated pipeline was proposed by Moon et al. for the abdominal spleen segmentation [26]. This pipeline offers an end-to-end synthesized process that eliminates the need for package installation while enabling local management of intermediate results through three major stages: pre-processing of input data, spleen segmentation based on SSNet using ResNet network and GAN, and 3D reconstruction by aligning the segmentation results with the original image dimensions for future use, display, or demonstration. The experimental results showed that the pipeline could provide fast and accurate segmentation of the spleen compared to traditional measurement methods.

Zettler et al. introduced a two-step approach for 3D segmentation on abdominal organs inside volumetric CT images, including liver, kidney, spleen, and pancreas [27]. In the approach, initially, a bounding box was generated to extract the volume of interest of each organ. This was then used as input for the second step involving segmentation using U-Nets with varying architectural dimensions. It was aimed at comparing the performance of 2D U-Nets against their 3D counterparts. According to the results, with a mean dice score of 0.93, 2D U-Net architecture showed promising results for 3D CT data.

Yuan et al. presented a framework based on a Variational Auto Encoder (VAE) to measure spleen volume from 2D spleen segmentations [28] was. Within this framework, three methods were proposed, and their performances for 3D CT data (both single- and dual-view) were evaluated. It was shown that the regression VAE connected layers with activation functions to model the potentially non-linear relationship between the latent embedding and volume (RVAE-FCNR) method achieved a mean relative volume agreement of 86.62% when using single-view data and 92.58% when using dual-view data.

The studies presented in the literature regarding the implementation of 2D CNN-based models for spleen segmentation from 3D CT scans are summarized in Table 1. As can be deduced from Table 1, while previous studies have proposed end-to-end pipelines and advanced frameworks for spleen segmentation, they often lack simplicity and generalizability required for deployment in clinical workflows.

On the other hand, while 2D CNN architectures offer significant advantages in terms of computational efficiency, their inherent limitation is the inability to fully capture inter-slice contextual information present in 3D medical images. This can lead to reduced segmentation accuracy, especially in anatomically complex regions or in cases where organ boundaries are ambiguous. To mitigate this limitation, hybrid strategies have been proposed, such as using adjacent slices as input channels or integrating context-aware attention modules [36–38]. However, these methods often introduce additional complexity that may conflict with the goal of achieving a lightweight and easily deployable solution [39].

Recent research has increasingly focused on model compression strategies to enable real-time segmentation on resource-constrained devices. Post-training quantization (PTQ) and pruning techniques have been shown to significantly reduce model size and memory footprint while maintaining accuracy within acceptable margins. For instance, Han et al. demonstrated that the combination of pruning and quantization can yield a compression rate up to 35× without accuracy degradation [40]. Similarly, EfficientQ proposes a fast PTQ method with a layer-wise ADMM optimization and self-adaptive attention for segmentation tasks, achieving significant memory reduction while preserving accuracy on BraTS and LiTS datasets [41]. Xu et al. applied both quantization-aware training and post-training quantization to fully convolutional networks (FCNs) for gland segmentation, reporting up to 6.4× memory reduction with even slight gains in segmentation accuracy [42]. In another study, evolutionary algorithms such as Differential Evolution (DE), Genetic Algorithms (GA), and Particle Swarm Optimization (PSO) were utilized to guide pruning for sparsity control in lung CT segmentation tasks, balancing model compactness and accuracy [43]. These approaches highlight the practical viability of compressed models

**Table 1. Studies on the use of 2D CNN-based models for spleen segmentation from 3D CT images.**

| Ref. | Methodology | Segmentation Approach | Key Findings |
|---|---|---|---|
| [26] | End-to-end pipeline with three stages: pre-processing, segmentation, and 3D reconstruction | SSNet with ResNet and GAN | Fast and accurate spleen segmentation |
| [27] | Two-step approach: bounding box extraction and segmentation | 2D and 3D U-Nets | 2D U-Net performed well with a mean Dice score of 0.93 |
| [28] | VAE framework with three proposed methods | RVAE-FCNR | Achieved 86.62% volume agreement (single-view) and 92.58% (dual-view) |
| Our work | 2D lightweight model processing the slice of 3D CT scans | U-Net based architecture using depthwise separable convolutions, model-aware pruning, and quantization | Reducing the computational burden while maintaining segmentation accuracy |

in clinical workflows, especially when integrated into lightweight CNN backbones. Nevertheless, although several studies have applied pruning and quantization techniques for model compression in medical image analysis, very few have addressed their integration into task-specific segmentation models, particularly for spleen CT scans.

As a summary, to the best of our knowledge, no existing study has introduced a lightweight, compression-aware 2D segmentation model specifically tailored for spleen segmentation from 3D CT scans. Therefore, this study proposes SVNC-Net, which combines architectural efficiency with post-training compression techniques to ensure optimal performance and deployability under real-world clinical conditions.

## The architecture of the SVNC-Net

The SVNC-Net is designed as an optimized and task-specific 2D variant of the traditional U-Net architecture, which is a widely used CNN for medical image segmentation [44]. The architecture of the model is shown in Fig 1. The model follows a symmetric encoder-decoder structure that enables efficient feature extraction while ensuring precise spatial reconstruction of segmented regions. It is designed to process grayscale or multi-channel medical images, such as CT scans, and generate segmentation masks that accurately define anatomical structures. The model accepts an input image of size 512×512 and outputs a segmentation map of the same spatial dimensions. In total, the architecture consists of 52 layers,

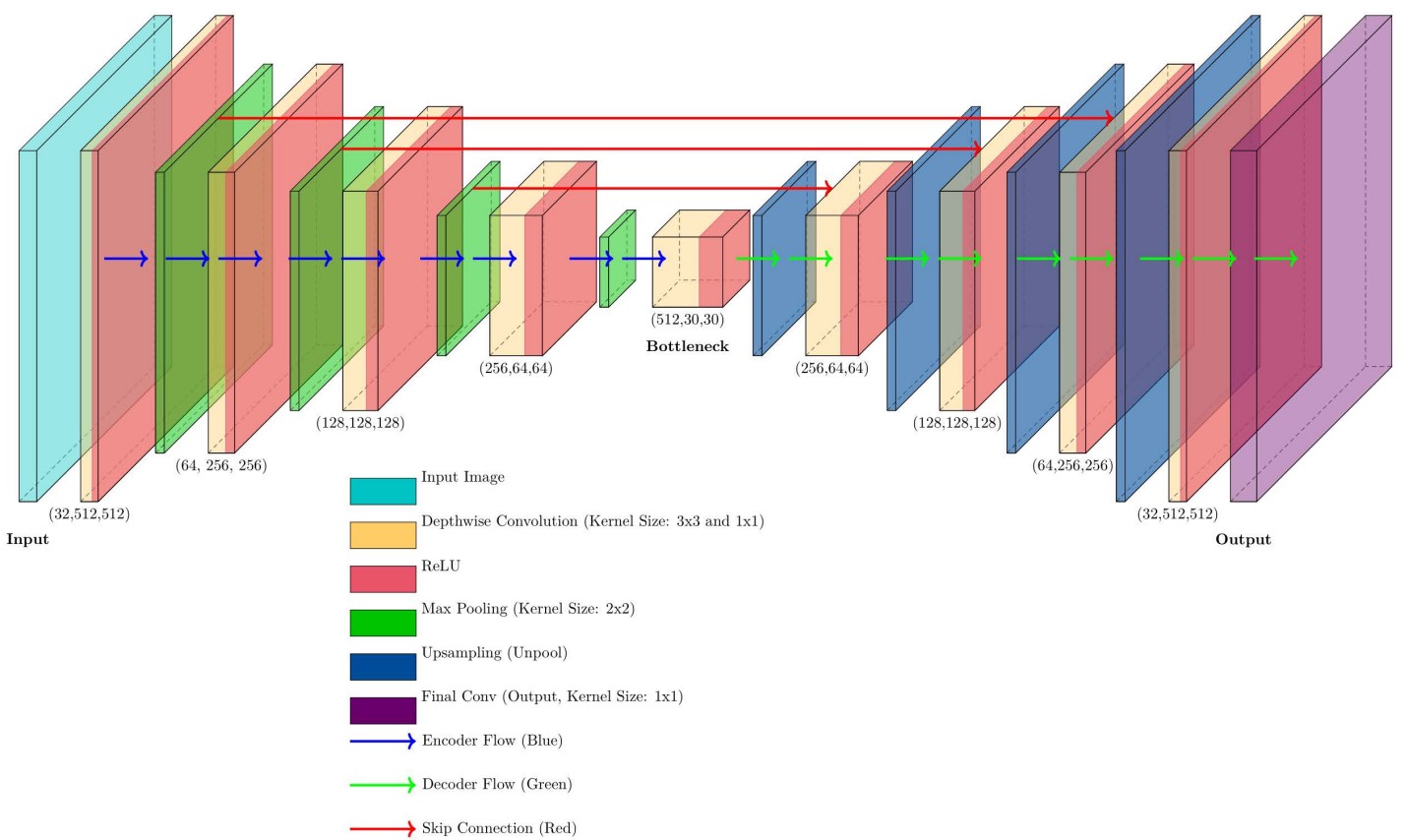

**Fig 1. The architecture of the SVNC-Net.**

incorporating convolutional layers, batch normalization, ReLU activation functions, pooling operations, and transposed convolutions for upsampling.

The encoder path comprises five downsampling stages, each consisting of convolutional operations followed by max-pooling layers that progressively reduce the spatial resolution while increasing the feature map depth. Initially, the input image of 512×512 pixels is processed by two standard convolutional layers with a kernel size of 3×3, producing 32 feature maps of the same resolution. These convolutions are followed by batch normalization and ReLU activation functions, which improve training stability and convergence. A 2×2 max-pooling layer subsequently reduces the spatial dimensions to 256×256, doubling the number of feature channels. The second stage introduces depthwise separable convolutions, which process 64 feature maps at a resolution of 256×256 before another max-pooling operation reduces the size to 128×128. The third stage further deepens the model by increasing the feature maps to 128, followed by max-pooling to 64×64. The fourth stage expands the feature map depth to 256, and downsampling reduces the resolution to 30×30. At the lowest resolution stage, known as the bottleneck, the network processes 512 feature maps at 30×30 pixels, capturing the highest level of abstraction and ensuring that the most representative features are learned.

At the bottleneck layer, the model applies additional depthwise separable convolutions and standard convolutions to refine high-dimensional feature representations. Here, as discussed by Howard et al., depthwise separable convolution is particularly efficient relative to standard convolution [45]. In Fig 2, the process of standard convolution and depthwise separable convolution is illustrated to demonstrate their structural differences. Specifically, the computational cost of standard convolutions is:

$$D_k \cdot D_k \cdot M \cdot N \cdot D_f \cdot D_f,$$
(1)

where it may depend multiplicatively on the number of input channels $M$, the number of output channels $N$, the feature map size $D_f \times D_f$, and the kernel size $D_k \times D_k$. Although the depthwise convolution has limited effectiveness with single-channel data, the 1×1 pointwise convolution efficiently adds new feature maps, ensuring an overall gain in performance. In this case, the computational cost of depthwise separable convolutions can be calculated by

$$D_k \cdot D_k \cdot M \cdot D_f \cdot D_f + M \cdot N \cdot D_f \cdot D_f.$$
(2)

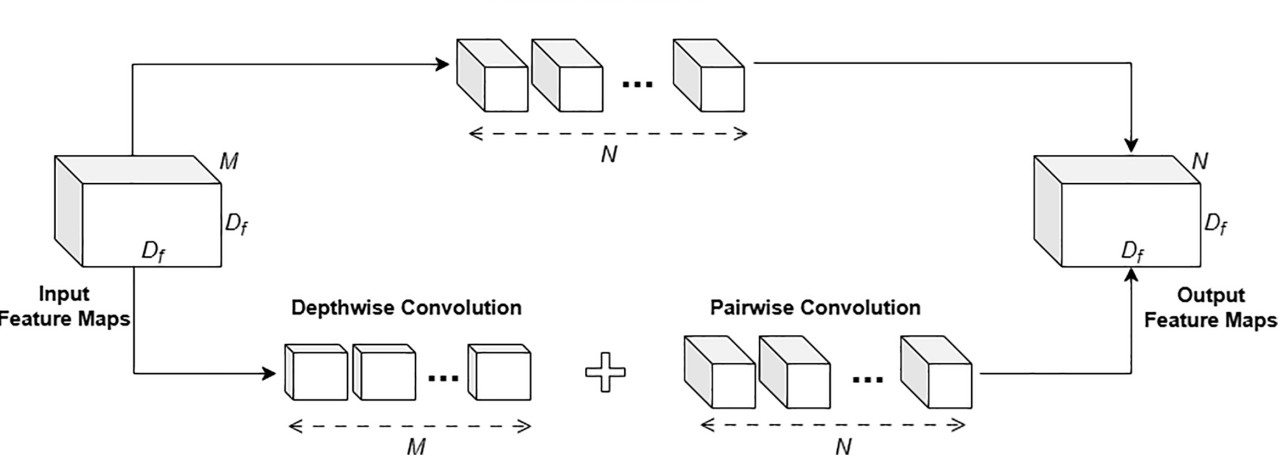

**Fig 2. The process of standard convolution and depthwise separable convolution.**

Then, decomposing the convolution operation into two distinct steps, namely filtering and combination, yields a reduction in computational complexity:

$$\frac{D_k \cdot D_k \cdot M \cdot D_f \cdot D_f + M \cdot N \cdot D_f \cdot D_f}{D_k \cdot D_k \cdot M \cdot N \cdot D_f \cdot D_f} = \frac{1}{N} + \frac{1}{D_k^2}.$$

(3)

This stage serves as a critical point in the network, ensuring that abstract spatial features are effectively learned before reconstruction begins. Once the feature extraction is complete, the decoder path restores the segmentation map by progressively upsampling the feature maps to their original resolution. The decoder consists of transposed convolutional layers that restore spatial dimensions while reducing feature map depth. Additionally, skip connections between corresponding encoder and decoder layers enable the network to retain fine-grained spatial details lost during downsampling. These connections help preserve anatomical boundaries and improve segmentation accuracy.

The upsampling process occurs in four stages. First, a transposed convolution increases the spatial resolution to 64×64, producing 256 feature maps. Skip connections are integrated from the corresponding encoder stage to fuse both high- and low-level spatial information. In the second upsampling stage, the resolution is increased to 128×128, with 128 feature maps being restored. The third upsampling stage further increases the resolution to 256×256, refining the segmentation details. Finally, in the fourth stage, the feature maps are upsampled to 512×512, with the number of feature channels reduced to 32. This ensures that the final segmentation mask maintains high spatial accuracy and is well-aligned with the input image.

The final prediction layer consists of a 1×1 convolution that outputs a segmentation mask of 512×512 pixels, ensuring that each pixel is classified into the appropriate category. The number of output channels in this layer depends on the segmentation task, with one output channel for binary segmentation and multiple channels for multi-class segmentation. The overall model is designed to optimize computational efficiency while maintaining high segmentation accuracy. Depthwise separable convolutions play a key role in reducing the number of trainable parameters while preserving feature extraction capacity [29]. Additionally, the inclusion of batch normalization and ReLU activation functions throughout the architecture ensures stable training dynamics and accelerates convergence.

The total number of layers in the model is 52, and its design is balanced between computational efficiency and segmentation performance. The use of skip connections helps maintain spatial integrity, while transposed convolutions ensure effective upsampling. The combination of standard and depthwise separable convolutions further enhances efficiency, making the model well-suited for large-scale medical imaging applications, such as spleen segmentation from CT scans. Given its structural design, the SVNC-Net is expected to perform effectively in clinical segmentation tasks, where accurate and computationally efficient solutions are required.

To provide a summary of the structural differences between the traditional U-Net and the SVNC-Net, the key architectural modifications and hyperparameter adjustments are presented in Table 2. The table outlines specific changes in convolutional operations, channel configurations, normalization strategies, and other core design elements, along with the motivation behind each modification.

## Experimental details

Experiments were conducted to assess the effectiveness of the SVNC-Net model and the well-known CNN-based models considered as baseline models using publicly available datasets. In the following sections, the baseline models are introduced, and then a brief overview of the publicly available datasets utilized for testing the models is provided. Additionally, the pre-processing steps and implementation details are described.

## Datasets

In the literature, only two single-organ segmentation datasets with CT scans for spleen are publicly available: the MSD dataset [34] and the DSDS [35]. In general, the MSD dataset is widely utilized in the field of medical image

**Table 2. Summary of key architectural modifications and hyperparameter adjustments from the U-Net to SVNC-Net.**

| Layer | U-Net | SVNC-Net | Motivation |
|---|---|---|---|
| Convolution Type | Standard 2D convolution | Depthwise separable | Reduce trainable parameters and computational complexity for mobile/edge deployment |
| Channel Numbers | 64–128–256–512 (bottleneck: 1024) | 32–64–128–256 (bottleneck: 512) | Prevent over-parameterization and reduce model size |
| Bottleneck Dilation | 1 | 2 | Expand receptive field without increasing parameters |
| Normalization | Optional batch normalization | 2D batch normalization in every block | Improve training stability and convergence speed |
| Upsampling | Transposed convolutions ($k=2$, $s=2$) | Transposed convolutions ($k=2$, $s=2$) and padding in skip connections | Eliminate fractional-stride artifacts |

segmentation as it provides a large volume and multi-layered imaging data for various segmentation tasks. It comprises 61 3D portal venous phase CT scans, among which 41 cases can be used for training purposes. A sample of MSD dataset is shown in Fig 3.

As an alternative to the MSD dataset, the DSDS has been provided to facilitate the development of spleen segmentation models. The dataset includes 109 anonymized CT and MRI volumes, comprising a total of 6322 images. Specifically, it is divided into 29 axial CT post-contrast series, 40 coronal SSFSE MRI series, and 40 axial opposed phase MRI series. Fig 4 shows a sample of the DSDS dataset.

Before performing the experiments, 3D CT images from the datasets were transformed into 2D slices that can be used in training phase. When the 41 CT scans in the MSD dataset were divided into slices, a total of 45634 images and 45634 corresponding masks were obtained. However, since a significant number of mask images contained no labels (images without a spleen), the entire dataset could not be used. Therefore, such unlabeled mask images and their corresponding scans were excluded from the dataset. Following this filtering process, 10750 images along with 10750 masks were retained. Furthermore, only slices along the z-axis were intended to be utilized. In fact, these slices represent cross-sectional layers that encode the depth information of the 3D volume, allowing for the reconstruction of the original 3D structure when combined. This approach preserves the spatial depth characteristics of the data, which is expected to enhance the effectiveness of the 2D segmentation model. Thus, with the use of slices along the z-axis only, a total of

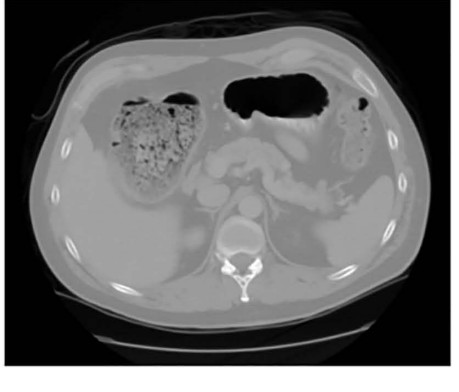 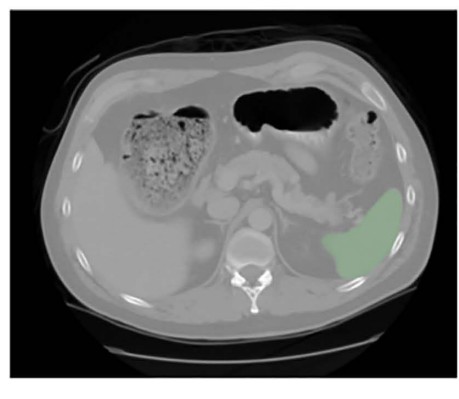 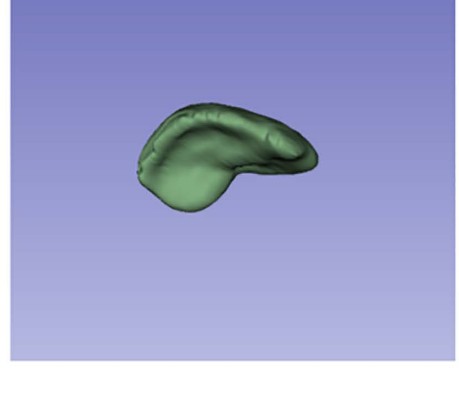

**(a)** **(b)** **(c)**

**Fig 3. A sample of MSD spleen dataset: (a) Raw image; (b) Ground truth (spleen region is indicated as green color); (c) 3D display.**

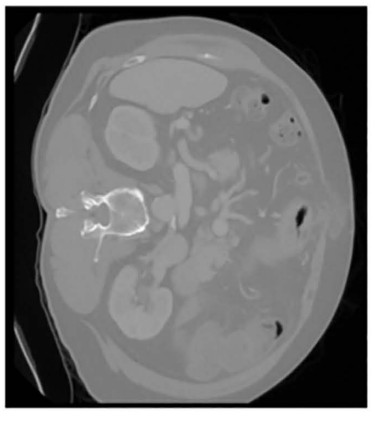 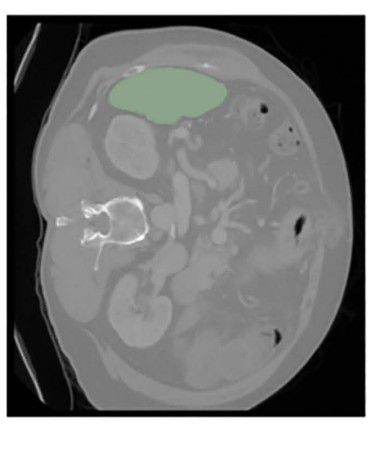 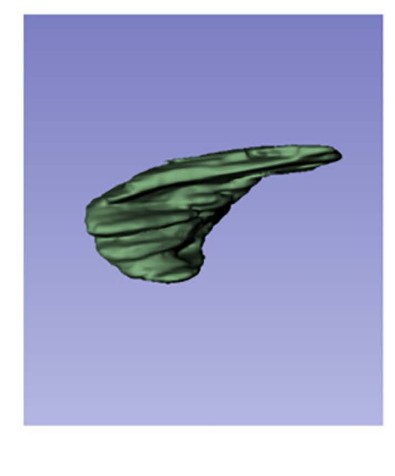

**Fig 4. A sample of DSDS spleen dataset: (a) Raw image; (b) Ground truth (spleen region is indicated as green color); (c) 3D display.**

1050 images and 1050 mask data were obtained. On the other hand, 29 CT (z-axis) scans in the DSDS could be used within the context of our study. After dividing them into slices, a total of 1000 images and 1000 corresponding masks were obtained. Before feeding the data into the models, the input dataset was resized to 512×512.

To ensure a reliable assessment of the performance of the SVNC-Net, both datasets were divided into 70% for training, 20% for validation, and 10% for testing. This distribution was chosen to effectively balance the evaluation of the generalization ability across different data subsets.

## Overview of the baseline models

In the experiments, the well-known and efficient CNN-based models in medical image segmentation, such as the UPerNet, EMANet, CCNet, SegNet, and ShuffleNet, were used as baseline models to verify the effectiveness of the SVNC-Net. Although these models have been implemented in various medical image segmentation tasks, their performances on spleen segmentation from CT images have not been reported yet. Therefore, we also aimed to assess their performances for the first time in the literature. In the following, the baseline models are briefly described.

**UPerNet.** UPerNet is a high-performance semantic segmentation model that integrates multi-scale feature representations through a Feature Pyramid Network (FPN) and a Pyramid Pooling Module (PPM) [29]. The architecture employs a deep convolutional backbone, such as ResNet-50, ResNet-101, or a transformer-based model like Swin Transformer, to extract multi-resolution feature maps. The FPN enhances feature propagation across scales, while the PPM captures global contextual dependencies by applying spatial pooling at multiple levels (e.g., 1×1, 2×2, 3×3, and 6×6). The decoder refines these representations through upsampling and convolutional operations to generate precise segmentation maps.

UPerNet-based models are effectively used in medical imaging for multi-organ segmentation, particularly in abdominal and thoracic imaging, liver segmentation, and brain tumor segmentation, using multi-scale feature aggregation [46–48].

**CCNet.** CCNet addresses the computational inefficiency of traditional self-attention mechanisms by introducing a Criss-Cross Attention (CCA) module, which captures contextual dependencies through a series of recurrent attention operations along horizontal and vertical directions [31]. The backbone, typically ResNet-50 or ResNet-101, extracts hierarchical feature representations, which are then processed by the CCA module. By iteratively aggregating context

from criss-cross paths, the model effectively captures long-range dependencies without the computational overhead of full self-attention.

CCA-based models are utilized in medical image segmentation for lung nodule segmentation, cardiac MRI segmentation, and real-time histopathology image segmentation due to its ability to model contextual dependencies and enhance boundary delineation [49,50].

**EMANet.** EMANet introduces an Expectation-Maximization (EM) attention mechanism to efficiently capture long-range dependencies in medical image segmentation [30]. The network utilizes a deep CNN backbone such as ResNet-50 or ResNet-101 to extract initial feature maps, which are then processed by the EM Attention Module. This module employs an iterative EM algorithm to learn a compact set of attention bases that summarize global contextual information. These bases are iteratively refined and used to generate attention coefficients that enhance feature representations while maintaining computational efficiency.

EM-based models are effective in medical imaging for segmenting heterogeneous structures like brain lesions and ischemic strokes, improving segmentation accuracy in complex organs like pancreatic tumors, and distinguishing between fine vessel structures and background noise [51–53].

**SegNet.** SegNet is an encoder-decoder-based segmentation model designed for efficient memory utilization and real-time inference [32]. The encoder follows the architecture of VGG-16, consisting of 13 convolutional layers with 3×3 kernels and ReLU activations. Spatial resolution is progressively reduced through five max-pooling layers, and the indices of these pooling operations are stored for later use in the decoder. Unlike conventional architectures, SegNet performs unpooling using these stored indices, allowing it to restore spatial resolution efficiently without the need for learned upsampling. The decoder consists of five unpooling layers followed by convolutional layers that refine the feature representations before classification.

The models based on SegNet are widely used in biomedical image segmentation tasks, particularly in lesion segmentation and histopathology image segmentation due to its ability to retain spatial features and process high-resolution microscopy data [54–56].

**ShuffleNet.** ShuffleNet is a computationally efficient deep learning model designed for high-speed inference on resource-constrained hardware, such as mobile and embedded systems [33]. The architecture incorporates pointwise group convolutions to reduce computational complexity, depthwise separable convolutions to enhance spatial feature extraction, and channel shuffle operations to improve information exchange between grouped features. The network consists of multiple ShuffleNet units, where each unit performs a series of grouped and depthwise convolutions, followed by channel shuffling to ensure effective feature propagation. The final stage of the network includes global average pooling and a fully connected layer for classification.

ShuffleNet-based models are used in medical image segmentation for real-time applications like lung ultrasound segmentation in COVID-19 diagnosis, and retinal fundus image segmentation [57,58].

## Training and implementation

In the training phase, the Adam optimizer was selected due to its robustness in handling the gradient instability commonly encountered in medical image segmentation tasks. In such problems, especially with organ segmentation from CT scans, some regions of the image contain rich anatomical information, while large areas may correspond to background with minimal signal. This imbalance often results in uneven gradient magnitudes across the network, with certain layers receiving weak gradients and others experiencing sharp updates. Adam optimizer offers a practical solution by incorporating estimates of both the first and second moments of the gradients to suppress noisy updates and sustain learning even in regions with low signal. Additionally, its adaptive learning rate mechanism allows each parameter to be updated at an appropriate scale, which is particularly beneficial in encoder-decoder architectures where the information density may vary significantly between layers.

As a result, Adam contributes to faster convergence and improved training stability in such heterogeneous data settings.

However, to achieve optimal performance, the learning rate was not fixed to a specific rate. Instead, a cosine annealing with warm restarts was employed to accelerate convergence [59]. In this approach, the learning rate ($\eta_t$) is dynamically adjusted at each iteration based on a cosine function:

$$\eta_t = \eta_{min} + \frac{1}{2}\left(\eta_{max} - \eta_{min}\right)\left(1 + \cos\left(\frac{T_c}{T_i}\pi\right)\right)$$

(4)

where $i$ is the index of the run, $T_i$ is the number of epochs between two warm restarts that defines the duration of each cosine annealing cycle, $T_c$ is the number of epochs since the last restart, $\eta_{min}$ is the minimum learning rate, and $\eta_{max}$ is set to the initial learning rate. In the training phase, then, $\eta_{max}$ was set to $1 \times 10^{-3}$, and $\eta_{min}$ was set to $1 \times 10^{-5}$. The values of $T_i$ were set to 10, 20, and 20 epochs for the first, second, and third cycles, respectively. Since the training of the models was completed in 50 epochs, the $T_i$ for the third cycle was set to 20.

To improve segmentation accuracy, the loss value was optimized using a combination of Dice Loss and Binary Cross-Entropy Loss. The combined loss ($L_{total}$) can be defined as [60]:

$$L_{total} = \alpha \cdot L_{Dice} + (1 - \alpha) \cdot L_{BCE}$$

(5)

where $L_{Dice}$ is the Dice Loss, $L_{BCE}$ is the Binary Cross-Entropy Loss, and $\alpha$ is the weighting factor that balances the contribution of the two loss components. To ensure that Dice Loss and Binary Cross-Entropy Loss contribute equally during training, the value of $\alpha$ was set to 0.5.

Moreover, a stratified partitioning strategy was adopted to ensure a reliable evaluation of the model's generalization capability while maintaining computational efficiency. Initially, the dataset was divided into 10 equal-sized, stratified folds ($F_1 - F_{10}$). Then, three independent evaluation rounds ($E_1 - E_3$) were designed, each following the standard 70% training, 20% validation, and 10% testing split. In each round, 7 folds were used for training, 2 for validation, and 1 for testing, as shown in Table 3. The main purpose of using this approach was to balance computational efficiency and statistical reliability. This approach is consistent with similar practices in the literature, where three to five evaluation rounds are often used [61,62]. While this approach does not correspond to full 10-fold cross-validation, which would require 10 independent training runs to preserve the same data proportions, it still allows for independent evaluation of the model across multiple test subsets. Importantly, not all folds were used as test sets. Instead, three distinct folds were selected for testing across the evaluation rounds, and the remaining folds were reused in different roles. It should also be noted that the validation sets were treated as internal evaluations used during training, while the test sets represent external, held-out evaluations used solely for final performance estimation. Although validation data were used during model development, no validation performance metrics were reported.

The experiments were conducted in a GPU environment provided by Google Colab that is optimized for processes requiring high computational power. More specifically, a single NVIDIA Tesla T4 GPU was used for the experiments. Moreover, the implementation was performed using the Python programming language with the PyTorch library.

As a summary, the training hyperparameters and implementation details discussed in this section is provided in Table 4.

**Table 3. Fold assignments for each evaluation round.**

| Evaluation Round | Training Folds | Validation Folds | Test Fold |
|---|---|---|---|
| $E_1$ | $F_1, F_3, F_4, F_6, F_7, F_8, F_{10}$ | $F_2, F_5$ | $F_9$ |
| $E_2$ | $F_2, F_4, F_5, F_6, F_7, F_9, F_{10}$ | $F_1, F_8$ | $F_3$ |
| $E_3$ | $F_1, F_2, F_3, F_5, F_7, F_8, F_9$ | $F_4, F_6$ | $F_{10}$ |

**Table 4. Training and implementation details.**

| Parameter/Component | Value |
|---|---|
| Optimizer | Adam |
| Initial Learning Rate | 0.001 |
| Learning Rate Scheduler | Cosine Annealing with Warm Restarts |
| Epochs | 50 |
| Batch Size | 16 |
| Loss Function | Dice Loss and Binary Cross-Entropy ($\alpha = 0.5$) |
| Data Splitting | 3 evaluation rounds using stratified 10-folds with 70% train/ 20% validation/ 10% test |
| Framework | Python, PyTorch |
| Hardware | Google Colab – NVIDIA Tesla T4 GPU |

## Post-Training compression

In addition to standard training and evaluation, this study investigates the impact of post-training model compression techniques, specifically pruning and quantization, on the SVNC-Net and baseline models. These techniques were applied only on models trained with the MSD dataset, to isolate computational effects from dataset-specific variances and avoid unnecessary complexity.

**Pruning strategy.** A model-aware pruning strategy was implemented to enable simplification while preserving the structural integrity of each network. Two pruning techniques were employed. The first one was weight pruning ($W$), which removes low-magnitude weights throughout the network to introduce sparsity with minimal disruption [63]. Another is neuron/filter pruning ($F$), which eliminates entire filters or neurons, resulting in a more substantial reduction in model complexity [64].

The pruning approach was tailored according to the architectural scale of each model. For more complex networks, including UPerNet, EMANet, CCNet, and SegNet, a combined strategy was applied, consisting of 15% weight pruning and 20% neuron/filter pruning ($[W + F]$). In contrast, for lightweight architectures such as SVNC-Net and ShuffleNet, only 15% weight pruning was employed to preserve the performance characteristics of these compact models. Importantly, no fine-tuning was conducted following pruning, allowing for a direct evaluation of the unrefined impact of sparsity on model performance.

**Quantization strategy.** PTQ was applied using 8-bit integer precision, converting model weights and activations from 32-bit floating point (float32) to 8-bit integer (int8) representations without additional retraining. This approach was adopted to emulate deployment scenarios on resource-constrained or embedded systems [65].

To assess the impact of both pruning and quantization, several evaluation metrics were reported, including post-training Intersection over Union (IoU), memory usage after compression, and inference time. These metrics served to quantify the trade-offs between segmentation accuracy and computational efficiency across the different model architectures.

## Results

This section is devoted to discussing the results obtained from the experiments. To this end, first, the metrics that were used for quantifying the performance of the models are outlined. Then, the effectiveness of the SVCN-Net and baseline models on the MSD dataset and DSDS is comparatively assessed.

## Performance metrics

To evaluate the efficiency of SVNC-Net and baseline models, it is essential to employ specific performance metrics that comprehensively assess their segmentation accuracy and computational demands. In the context of segmentation

accuracy, key evaluation metrics, including IoU, mean IoU (mIoU), and the Dice Similarity Coefficient (DSC), were utilized. Additionally, computational efficiency was assessed using metrics, such as inference time ($t_{inf}$), total memory usage (MemU), training time ($t_{train}$), and the number of model parameters ($N_{params}$). A brief explanation of each metric is presented in the following sections.

### Intersection over union (*IoU*)

The *IoU* metric is a generalization of the Jaccard index which is widely used for evaluating the accuracy of the segmentation algorithms [66]. It measures the overlap between the predicted segmentation region and the ground truth segmentation region in medical images, such as organs, tumors, or lesions. It can be calculated as the predicted segmentation (*A*) and the ground truth segmentation (*B*):

$$IoU = \frac{|A \cap B|}{|A \cup B|},$$

(6)

where $|A \cap B|$ denotes the number of pixels shared between the predicted and ground truth masks, and $|A \cup B|$ denotes the total number of pixels in either the predicted or ground truth masks. The *IoU* ranges from 0 to 1, inclusive. Higher values indicate higher segmentation performance.

### Mean intersection over union (*mIoU*)

The m*IoU* is the average *IoU* for all classes in a given dataset. It can be expressed by averaging the *IoU* values for all segmentation classes:

$$mIoU = \frac{1}{n} \sum_{i=1}^{n} IoU_i.$$

(7)

where *n* is the total number of classes, and $IoU_i$ is the *IoU* for the *i*th class that can be defined as

$$IoU_i = \frac{|A_i \cap B_i|}{|A_i \cup B_i|},$$

(8)

where $A_i$ is the predicted segmentation for the *i*th class, and $B_i$ is the ground truth segmentation for the *i*th class.

The m*IoU* metric is particularly important in multi-class segmentation task. Nevertheless, in this study, m*IoU* was computed by averaging the *IoU* values of the spleen (foreground) and background classes to better reflect overall segmentation performance. In fact, the m*IoU* metric was adopted by calculating the *IoU* separately for both classes and then averaging the results. This approach aligns with the pixel-wise classification nature of segmentation tasks, where both foreground and background predictions are important. Including background performance is particularly useful for evaluating how well the model balances class predictions, especially in cases with significant class imbalance [67].

### Dice similarity coefficient (*DSC*)

The *DSC* is a spatial overlap index proposed by Dice [68] used to measure measures the overlap between the predicted and ground truth segmentations. It can be defined as follows

$$DSC = \frac{2|A \cap B|}{|A| + |B|}.$$

(9)

As it is often used alongside *IoU*, it can also be defined as follows

$$DSC = \frac{2 \times IoU}{1 + IoU}.$$

(10)

The DSC ranges from 0 to 1, where the value of one indicates the highest segmentation accuracy, and lower values indicate significant deviations between the predictions and the ground truth mask.

## Computational efficiency metrics

In clinical applications, particularly medical image segmentation, the selection of optimal DL models requires a comprehensive evaluation of computational cost and scalability. Key indicators such as inference time ($t_{inf}$), total memory usage (MemU), training time ($t_{train}$), and the number of model parameters ($N_{params}$) may directly affect the feasibility and efficiency of deployment. Therefore, in this study, these metrics were utilized to provide alternative assessments of model efficiency beyond segmentation accuracy.

Rapid inference is important for real-time diagnostics and efficient clinical workflows, while minimized memory usage facilitates deployment on resource-constrained medical devices and reduces operational costs. Furthermore, expedited training times enable rapid model development, adaptation to evolving clinical data, and accelerated research. An acceptable balance in model parameterization ensures robust generalization, mitigates overfitting, and optimizes computational resource utilization, thereby enhancing the reliability and applicability of AI-driven medical image analysis.

The significance of these metrics becomes more prevalent in medical image segmentation due to the complex nature of medical imagery and the critical need for accurate and timely analyses. In clinical practice, where decisions are often time-sensitive and resource limitations are tight, models should demonstrate both high accuracy and computational efficiency. Thus, the selection process requires a comprehensive assessment of these factors to ensure that the chosen model not only meets the diagnostic requirements but also aligns with the practical constraints of the clinical environment.

## Segmentation accuracy

The segmentation accuracy of SVNC-Net and baseline models on the MSD and DSDS datasets, obtained from the 10% testing subset, is presented in Table 5. As can be seen from the table, SVNC-Net achieved an IoU of 0.89, an mIoU of 0.94, and a DSC of 0.94 on the MSD dataset. This indicates that SVNC-Net performed competitively, matching the performance of significantly larger models such as UPerNet, which achieves an IoU of 0.87, an mIoU of 0.93, and a DSC of 0.93, EMANet achieved an IoU of 0.89, an mIoU of 0.95, and a DSC of 0.94, and CCNet had an IoU of 0.91, an mIoU of 0.95, and a DSC of 0.95. Therefore, it can be deduced that SVNC-Net is able to achieve comparable segmentation accuracy on the MSD dataset.

**Table 5. Segmentation accuracy of the models.**

| Model | MSD | | | DSDS | | |
|---|---|---|---|---|---|---|
| | IoU | mIoU | DSC | IoU | mIoU | DSC |
| UPerNet | 0.87 | 0.93 | 0.93 | 0.83 | 0.91 | 0.91 |
| CCNet | 0.91 | 0.95 | 0.95 | 0.76 | 0.88 | 0.86 |
| EMANet | 0.89 | 0.95 | 0.94 | 0.76 | 0.87 | 0.86 |
| SegNet | 0.82 | 0.91 | 0.90 | 0.71 | 0.85 | 0.83 |
| ShuffleNet | 0.84 | 0.92 | 0.91 | 0.83 | 0.91 | 0.91 |
| SVNC-Net | 0.89 | 0.94 | 0.94 | 0.84 | 0.92 | 0.92 |

On the DSDS dataset, with an IoU of 0.84, an mIoU of 0.92, and a DSC of 0.92, SVNC-Net achieved the highest performance when compared to baseline models. Although these results are comparable to those of ShuffleNet and UPerNet, both of which achieved an IoU of 0.83, an mIoU of 0.91, and a DSC of 0.91, SVNC-Net still maintained a high accuracy. The ability of SVNC-Net to generalize across multiple datasets further reinforces its applicability in real-world scenarios.

On the other hand, the performance analysis of the models across the MSD dataset and DSDS reveals notable variations. EMANet and CCNet demonstrated better performance on the MSD dataset. However, their performance significantly reduced on the DSDS dataset. This may indicate a lack of generalizability across different datasets. Moreover, SegNet demonstrated poor performance on both datasets, which may raise concerns about its limited effectiveness in segmentation tasks. UPerNet, while performing slightly worse on DSDS in comparison to MSD, maintained a relatively stable and acceptable performance compared to other baseline models. Furthermore, the performance of ShuffleNet was robust, which achieved nearly identical performance across both datasets. Nevertheless, when considering overall results, SVNC-Net outperformed ShuffleNet, particularly when its lightweight architecture is taken into account.

## Computational performance

Computational efficiency of the models are provided in Table 6. It is important to note that this table presents the results obtained from the MSD dataset only, as similar results were observed on the DSDS dataset. From the table, the lightweight nature of SVNC-Net can be clearly observed. As listed in the table, it contains only 1 million (M) parameters ($N_{params}$), which is substantially smaller than other efficient models, such as UPerNet (60.1M) and ShuffleNet (5.5M). Obviously, SVNC-Net is the most compact model that highlights its design efficiency.

As discussed earlier, inference time ($t_{inf}$) is another important metric to select a proper model for clinical applications. According to the results, SVNC-Net demonstrated the lowest inference time (34.4 ms). It is significantly faster than other models, even than relatively lightweight ShuffleNet (40.3 ms). This is particularly beneficial for real-time applications, where timely segmentation results are critical.

Total memory usage (MemU) is another defining characteristic of SVNC-Net. Requiring only 4MB of memory, it is more efficient than other efficient models such as UPerNet (253MB) and ShuffleNet (21MB). This makes SVNC-Net highly suitable for deployment in resource-constrained environments, such as embedded systems or clinical workstations with limited computational capacity.

Training efficiency is another advantage of SVNC-Net. With a training time ($t_{train}$) of only 29 min, it significantly outperformed larger models, such as UPerNet (179 min) and CCNet (162 min), which require substantially more computational resources. Particularly, it had lower training time in comparison to its lightweight rival ShuffleNet (41 min). In clinical environments, the reduced training time not only enhances practicality but also facilitates rapid model updates.

## Compression evaluation

The impact of compression techniques on segmentation models trained on the MSD dataset is summarized in Table 7. In general, pruning (both weighted and filter-based) yielded consistent reductions in parameter count and inference latency,

**Table 6. Computational efficiency of the models.**

| Model | $N_{params}$ (M) | $t_{inf}$ (ms) | MemU (MB) | $t_{train}$ (min) |
|---|---|---|---|---|
| UPerNet | 60.1 | 216 | 253 | 179 |
| CCNet | 49.8 | 236 | 190 | 162 |
| EMANet | 42 | 184 | 161 | 134 |
| SegNet | 24.7 | 132.6 | 94 | 89 |
| ShuffleNet | 5.5 | 40.3 | 21 | 41 |
| SVNC-Net | 1 | 34.4 | 4 | 29 |

**Table 7. Impacts of pruning and quantization on the model performances.**

| Model | IoU | $N_{params}$* | $t_{inf}$ (ms) | MemU (MB) |
|---|---|---|---|---|
| UPerNet | 0.87 | 60.1 | 216 | 253 |
| UPerNet-Pruned | 0.86 | 40.8 | 154 | 180 |
| UPerNet-Quantized | 0.83 | 60.1 | 194 | 164 |
| CCNet | 0.91 | 49.8 | 236 | 190 |
| CCNet-Pruned | 0.88 | 33.8 | 171 | 142 |
| CCNet-Quantized | 0.89 | 49.8 | 212.4 | 124 |
| EMANet | 0.89 | 42 | 184 | 161 |
| EMANet-Pruned | 0.88 | 28.6 | 140 | 115 |
| EMANet-Quantized | 0.88 | 42 | 165.6 | 105 |
| SegNet | 0.82 | 24.7 | 132.6 | 94 |
| SegNet-Pruned | 0.77 | 16.8 | 92 | 67 |
| SegNet-Quantized | 0.81 | 24.7 | 119 | 63 |
| ShuffleNet | 0.84 | 5.5 | 40.3 | 21 |
| ShuffleNet-Pruned | 0.83 | 4.7 | 38 | 20.4 |
| ShuffleNet-Quantized | 0.81 | 5.5 | 36.3 | 17 |
| SVNC-Net | 0.89 | 1 | 34.4 | 4 |
| SVNC-Net-Pruned | 0.88 | 0.9 | 33 | 3.8 |
| SVNC-Net-Quantized | 0.86 | 1 | 30 | 3 |

*$N_{params}$ are exact values rounded to one decimal place in millions (M); no averaging was applied.

while quantization substantially reduced memory consumption. However, the magnitude of performance degradation varied depending on model architecture and initial complexity. For heavyweight models such as UPerNet and CCNet, pruning led to significant reductions in parameters and faster inference, while only slightly decreasing IoU, from 0.87 to 0.86 for UPerNet and from 0.91 to 0.88 for CCNet. These models, which involve complex multi-branch structures or attention modules, benefit from structured pruning, though quantization appears more challenging, especially for UPerNet, which includes LayerNorm and deeply nested operations known to be sensitive to low-precision computations. While UPerNet's quantized variant experienced a more pronounced drop in IoU (to 0.83), CCNet exhibited strong robustness, maintaining an IoU of 0.89 after quantization while achieving a 34.7% reduction in memory usage. This may be due to its well-isolated Criss-Cross attention units, which are less susceptible to quantization error propagation. Similarly, EMANet demonstrated excellent compression tolerance, with both pruning and quantization preserving an IoU of 0.88 while reducing memory requirements. This indicates the relative resilience of EM-based attention structures to weight perturbations caused by quantization. In contrast, SegNet displayed heightened sensitivity to pruning, with a significant drop in IoU from 0.82 to 0.77, which can be attributed to its shallow encoder-decoder design lacking residual or normalization pathways to compensate for parameter loss. Interestingly, the quantized variant partially recovered accuracy, reaching an IoU of 0.81, which suggests that quantization, especially when applied to simpler architectures, may be a less destructive compression strategy than aggressive filter pruning.

In the lightweight category, both ShuffleNet and the proposed SVNC-Net exhibited strong compression resilience and real-time feasibility. ShuffleNet maintained competitive IoU values (0.825 after pruning, 0.81 after quantization), with memory usage reduced to just 17 MB. However, this slight drop in accuracy can be attributed to its reliance on grouped and channel-shuffled convolutions, which are structurally more vulnerable to 8-bit quantization errors due to reduced redundancy. Notably, SVNC-Net achieved the best balance between accuracy and efficiency, starting with an IoU of 0.89 and maintaining 0.882 and 0.86 after pruning and quantization, respectively. Despite incorporating depthwise separable

convolutions, which are known to be sensitive to low-precision arithmetic, SVNC-Net preserved stable performance, suggesting that its layerwise placement of depthwise separable convolutions effectively mitigates quantization artifacts. With only 1M parameters and a final memory footprint as low as 3 MB, the quantized SVNC-Net proved to be the most lightweight model tested. These findings strongly support the suitability of SVNC-Net for deployment in real-time, edge-based medical image segmentation systems, where both accuracy and efficiency are critical.

### Qualitative evaluation

To visually assess the segmentation performance of the proposed SVNC-Net and other baseline models, qualitative comparisons were conducted on MSD and DSDS. Ten representative CT slices, comprising five from each dataset, were selected to illustrate performance differences, particularly in terms of boundary delineation and regional consistency. The results are shown in Figs 5 and 6. In all visualization samples, the ground truth is overlaid as a red boundary line, while the predicted segmentation masks are shown as filled regions for each model. This enables clear visual comparison of boundary alignment and segmentation consistency across different methods. All segmentations were rendered using the same color scheme to maintain visual uniformity. As can be seen from the figures, across both datasets, SVNC-Net displays more precise boundary localization and better region consistency, particularly when compared to ShuffleNet and SegNet. Although UPerNet and CCNet also perform well, their computational overhead is significantly higher. SVNC-Net achieves a competitive visual quality while maintaining lightweight and real-time feasibility, confirming its suitability for edge-based clinical applications.

## Discussion

The results obtained from the experiments validate that SVNC-Net is a lightweight, efficient, and clinically applicable spleen segmentation model from 3D CT scans. To the best of our knowledge, no prior model has been specifically designed to achieve such an optimal balance between segmentation accuracy and computational efficiency. SVNC-Net's ability to maintain high segmentation performance while minimizing computational costs establishes its suitability for real-world clinical deployment.

### Trade-off between lightweight and continuity

SVNC-Net presents an optimal trade-off between segmentation accuracy and computational efficiency. Its lightweight structure ensures ease of deployment, while its low inference time and memory consumption make it ideal for real-time applications. The combination of high accuracy and computational efficiency highlights its potential for spleen segmentation in clinical settings, where timely and reliable segmentation is essential for diagnostic and treatment planning. Given these advantages, SVNC-Net represents a significant advancement in spleen segmentation from CT scans, providing a robust solution that meets the demands of modern clinical workflows.

SVNC-Net operates exclusively on 2D slices, processing 3D data by converting it into individual 2D sections. This approach inherently leads to the potential loss of spatial contextual information across the x, y, and z axes. Since 3D structures rely on inter-slice dependencies, this constraint could affect the ability to fully capture volumetric relationships. However, despite this theoretical limitation, the model remains highly effective in 2D segmentation tasks. Its primary focus is on extracting features from individual slices, and its performance does not show a significant decline due to the absence of explicit 3D spatial reasoning. The architecture of SVNC-Net is well-optimized for its intended application, demonstrating strong segmentation capabilities even without direct volumetric awareness.

One key area where the lack of 3D context could affect performance is in object boundary definition. In cases where boundaries are ambiguous, additional spatial cues from adjacent slices could enhance segmentation accuracy. Without such information, SVNC-Net might face challenges in distinguishing closely positioned or overlapping structures. However, the lightweight

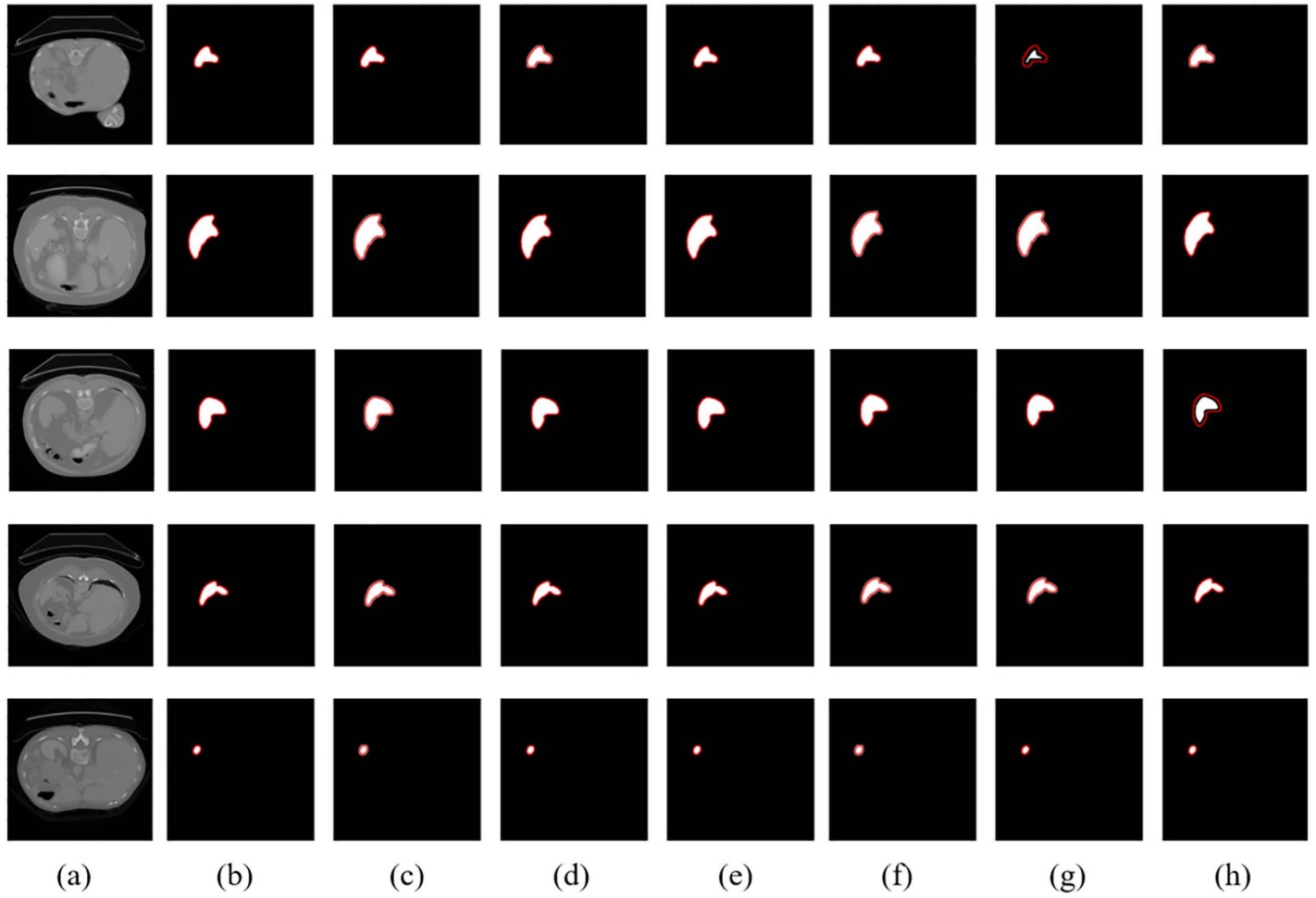

**Fig 5. Qualitative comparison of spleen segmentation results on the MSD dataset: (a) Input CT slice, (b) Ground Truth (red boundary), (c) UPerNet prediction, (d) EMANet prediction, (e) CCNet prediction, (f) SegNet prediction, (g) ShuffleNet prediction, (h) SVNC-Net prediction.** The red contours represent the expert-annotated ground truth masks.

and simplistic architecture of the model minimizes the impact of this limitation. Unlike models that rely on highly rigid feature learning, SVNC-Net maintains a more flexible representation, reducing the risk of structural overfitting. This characteristic allows the model to be less sensitive to the absence of 3D contextual information, mitigating the potential drawbacks of its 2D-based approach.

Therefore, while the absence of direct 3D spatial reasoning introduces constraints, SVNC-Net continues to deliver robust performance in 2D segmentation tasks. The balance between computational efficiency and segmentation accuracy remains favorable, making it well-suited for applications where computational cost is prioritized over full volumetric understanding.

## Post-hoc compression and clinical feasibility

In addition to its native efficiency, SVNC-Net benefits further from post-hoc compression techniques. As discussed earlier, pruning and 8-bit PTQ were applied to reduce model size, latency, and memory usage, with minimal impact on segmentation accuracy. These methods offer practical gains in edge-based clinical deployment, where hardware constraints are critical.

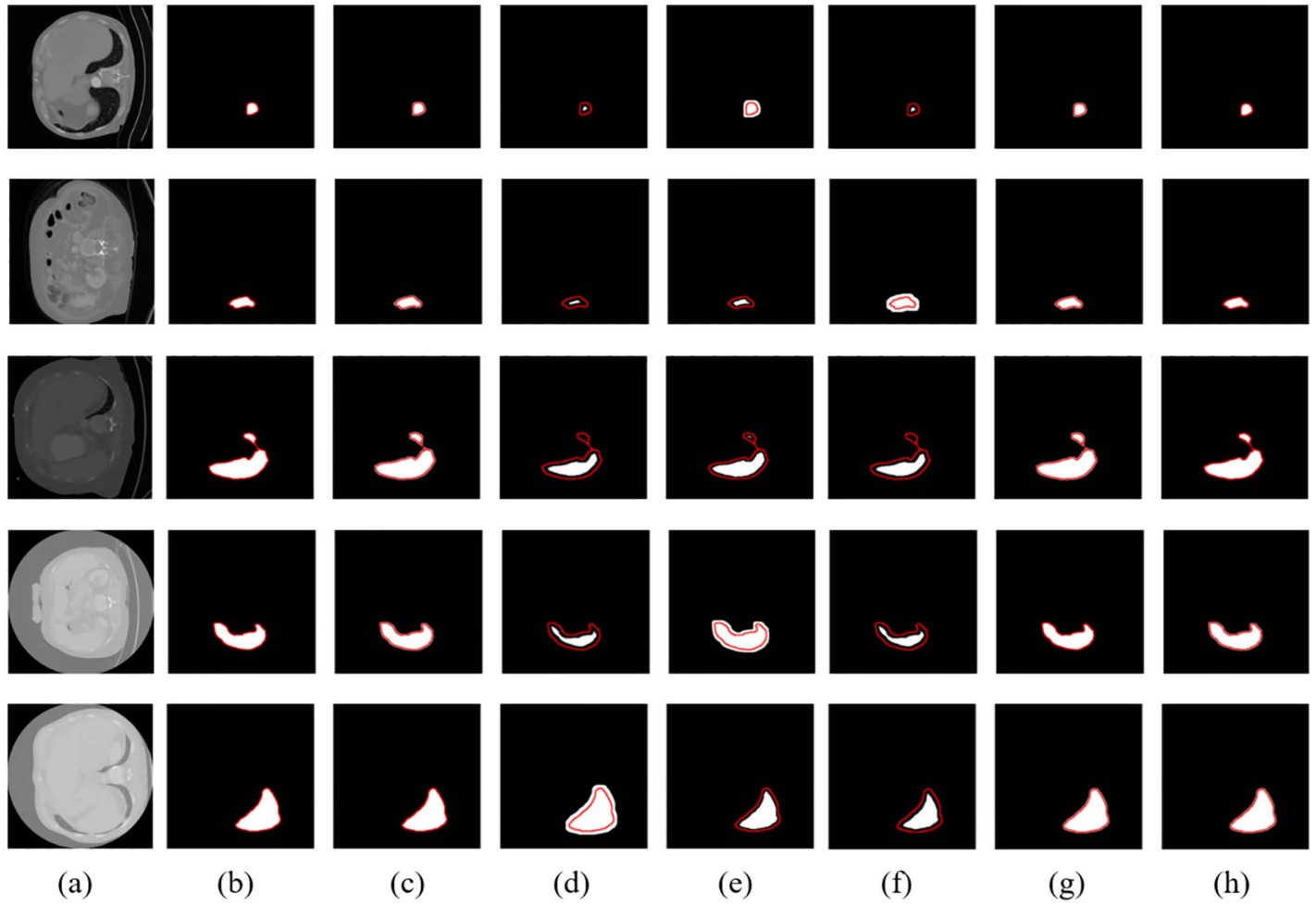

**Fig 6. Qualitative comparison of spleen segmentation results on the DSDS: (a) Input CT slice, (b) Ground Truth (red boundary), (c) UPerNet prediction, (d) EMANet prediction, (e) CCNet prediction, (f) SegNet prediction, (g) ShuffleNet prediction, (h) SVNC-Net prediction.** The red contours represent the expert-annotated ground truth masks.

Notably, weight pruning was selectively applied to lightweight models such as ShuffleNet and SVNC-Net, rather than filter-based pruning. This decision stems from the architectural sensitivity of such models: aggressive structural pruning may distort spatial filters due to their already minimal redundancy.

The compression results can be summarized as follows:

- The memory usage of SVNC-Net was reduced by 25% through quantization (from 4MB to 3MB), with only a 3.4% drop in IoU.

- Inference time decreased from 34.4 ms to 30 ms via quantization, enhancing real-time capability.

- Parameter count was reduced by 10% through weight pruning, with only a 0.9% performance drop.

These outcomes validate that SVNC-Net not only performs well in its original configuration but is also amenable to further compression without violating performance thresholds essential for clinical reliability.

### Clinical implications of segmentation accuracy

While computational efficiency remains a primary motivation for the development of SVNC-Net, its segmentation accuracy also holds meaningful clinical relevance. In clinical settings, even marginal improvements in segmentation accuracy can lead to more reliable estimations of organ volume, which are crucial for diagnosing conditions such as splenomegaly. Particularly, in borderline cases where the spleen volume is close to the diagnostic threshold, a slight increase in accuracy metrics, such as *IoU* or *DSC*, may reduce the risk of misdiagnosis or unnecessary follow-up procedures.

Moreover, SVNC-Net consistently demonstrated high segmentation performance across two distinct datasets, despite their differences in imaging protocols and population characteristics. This stability may suggest a high degree of robustness and generalizability, which are essential for deployment in real-world clinical environments where variations in scan quality, acquisition protocols, and patient anatomy are frequently encountered.

Although some baseline models demonstrated comparable segmentation accuracies, SVNC-Net achieved this performance with significantly lower memory usage, faster inference, and greater robustness to compression. Importantly, this balance of accuracy, efficiency, and resilience is particularly valuable for deployment on edge devices in point-of-care settings, where limited computational resources coincide with the need for reliable and timely predictions. By maintaining competitive accuracy while significantly reducing memory usage and inference time, SVNC-Net offers a clinically viable solution for automated spleen segmentation in time-sensitive and resource-constrained clinical workflows.

## Conclusion

This study proposes SVNC-Net, which is a lightweight 2D CNN-based model for efficient 3D spleen segmentation from CT scans. SVNC-Net employs depthwise separable convolutions to enhance computational efficiency while maintaining high segmentation accuracy. Evaluated on the MSD and DSDS, SVNC-Net achieved competitive segmentation accuracy, with an IoU of 0.89 and DSC of 0.94 on MSD, and an IoU of 0.84 and DSC of 0.92 on DSDS. Additionally, SVNC-Net outperformed well-known CNN-based models in computational efficiency, with only 1M parameters, the lowest inference time (34.4 ms), and minimal memory usage (4 MB). With a training time of 29 minutes, SVNC-Net is highly suitable for real-time clinical applications, offering a promising approach for lightweight 2D deep learning models in 3D organ segmentation. Furthermore, post hoc compression results revealed that quantization reduced memory usage by 25% (from 4 MB to 3 MB) and inference time by 12.8%, with only a 3.4% drop in IoU, demonstrating the model's robustness under hardware friendly optimizations.

In conclusion, this study demonstrates that SVNC-Net is a computationally efficient and highly accurate model for 3D spleen segmentation from CT scans. Using a lightweight 2D architecture, it significantly reduces the computational burden while maintaining segmentation accuracy comparable to larger models. The higher performance of SVNC-Net in terms of inference speed and memory usage highlights its potential for integration into clinical workflows, particularly in resource-constrained environments. In addition, the successful application of pruning and quantization techniques further demonstrates the potential of SVNC-Net for real-time deployment on edge medical devices. Future research can explore its application to other organ segmentation tasks and further optimize its architecture for broader medical imaging applications.

## Author contributions

**Conceptualization:** Mehmet Zahid Genc, Yaser Dalveren.

**Formal analysis:** Mehmet Zahid Genc, Yaser Dalveren.

**Funding acquisition:** Marek Penhaker.

**Investigation:** Mehmet Zahid Genc, Yaser Dalveren.

**Methodology:** Yaser Dalveren.

**Project administration:** Marek Penhaker.

**Software:** Mehmet Zahid Genc.

**Supervision:** Ali Kara, Mohammad Derawi, Jan Kubicek, Marek Penhaker.

**Validation:** Ali Kara, Mohammad Derawi, Jan Kubicek.

**Writing – original draft:** Mehmet Zahid Genc, Yaser Dalveren.

**Writing – review & editing:** Ali Kara, Mohammad Derawi, Jan Kubicek.

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
