## [Decision Letter · Decision Letter 0]

2 Jul 2025

Dear Dr. Dalveren,

Thank you for submitting your manuscript to PLOS ONE. After careful consideration, we feel that it has merit but does not fully meet PLOS ONE’s publication criteria as it currently stands. Therefore, we invite you to submit a revised version of the manuscript that addresses the points raised during the review process.

We look forward to receiving your revised manuscript.

Kind regards,

Marco Antonio Moreno-Armendariz, Ph.D.

Academic Editor

PLOS ONE

**Journal Requirements:**

1. When submitting your revision, we need you to address these additional requirements. Please ensure that your manuscript meets PLOS ONE's style requirements, including those for file naming. The PLOS ONE style templates can be found at https://journals.plos.org/plosone/s/file?id=wjVg/PLOSOne_formatting_sample_main_body.pdf and https://journals.plos.org/plosone/s/file?id=ba62/PLOSOne_formatting_sample_title_authors_affiliations.pdf 2. Please note that PLOS ONE has specific guidelines on code sharing for submissions in which author-generated code underpins the findings in the manuscript. In these cases, we expect all author-generated code to be made available without restrictions upon publication of the work. Please review our guidelines at https://journals.plos.org/plosone/s/materials-and-software-sharing#loc-sharing-code and ensure that your code is shared in a way that follows best practice and facilitates reproducibility and reuse. 3. Thank you for stating in your Funding Statement: This article has been produced with the financial support of the European Union under the LERCO project number CZ.10.03.01/00/22_003/0000003 via the Operational Programme Just Transition. The work and the contributions were supported by the project SP2025/032 ‘Biomedical Engineering systems XXI’.   Please provide an amended statement that declares *all* the funding or sources of support (whether external or internal to your organization) received during this study, as detailed online in our guide for authors at http://journals.plos.org/plosone/s/submit-now.  Please also include the statement “There was no additional external funding received for this study.” in your updated Funding Statement. Please include your amended Funding Statement within your cover letter. We will change the online submission form on your behalf. 4. Thank you for stating the following in the Acknowledgments Section of your manuscript: This article has been produced with the financial support of the European Union under the LERCO project number CZ.10.03.01/00/22_003/0000003 via the Operational Programme Just Transition. The work and the contributions were supported by the project SP2025/032 ‘Biomedical Engineering systems XXI’. We note that you have provided funding information that is not currently declared in your Funding Statement. However, funding information should not appear in the Acknowledgments section or other areas of your manuscript. We will only publish funding information present in the Funding Statement section of the online submission form. Please remove any funding-related text from the manuscript and let us know how you would like to update your Funding Statement. Currently, your Funding Statement reads as follows: This article has been produced with the financial support of the European Union under the LERCO project number CZ.10.03.01/00/22_003/0000003 via the Operational Programme Just Transition. The work and the contributions were supported by the project SP2025/032 ‘Biomedical Engineering systems XXI’.   Please include your amended statements within your cover letter; we will change the online submission form on your behalf. 5. Thank you for uploading your study's underlying data set. Unfortunately, the repository you have noted in your Data Availability statement does not qualify as an acceptable data repository according to PLOS's standards. At this time, please upload the minimal data set necessary to replicate your study's findings to a stable, public repository (such as figshare or Dryad) and provide us with the relevant URLs, DOIs, or accession numbers that may be used to access these data. For a list of recommended repositories and additional information on PLOS standards for data deposition, please see https://journals.plos.org/plosone/s/recommended-repositories.

**Additional Editor Comments:**

Please consider the comments by the reviewers.

Reviewers' comments:

Reviewer's Responses to Questions

**Comments to the Author**

1. Is the manuscript technically sound, and do the data support the conclusions?

Reviewer #1: Yes

Reviewer #2: Partly

2. Has the statistical analysis been performed appropriately and rigorously?

Reviewer #1: Yes

Reviewer #2: N/A

3. Have the authors made all data underlying the findings in their manuscript fully available?

Reviewer #1: Yes

Reviewer #2: Yes

4. Is the manuscript presented in an intelligible fashion and written in standard English?

Reviewer #1: Yes

Reviewer #2: Yes

**Reviewer #1:**  The authors present an optimized version of the U-Net architecture for performing semantic segmentation on spleen images, achieving improvements in both memory usage and training/inference time. However, a more detailed comparison is needed to clarify which hyperparameters were modified during the optimization process.

The authors include both numerical and visual comparisons of their results; nevertheless, several improvements are recommended for the diagrams presented. First, Figure 1 requires more detailed information: the kernel size used in each convolution and transposed convolution should be specified, as well as the spatial dimensions of the feature maps after each convolutional and deconvolutional operation. Additionally, the kernel size used for max pooling should be indicated.

It would also be highly beneficial to include a diagram comparing the original U-Net architecture with the proposed optimized version, clearly highlighting the modifications made. Furthermore, a figure illustrating the difference between standard convolution and depthwise separable convolution should be added.

For Figures 4 and 5, it is recommended to label each column with a corresponding letter to make it easier to interpret the results of each experiment.

**Reviewer #2: ** In this manuscript, authors introduce a novel architecture for spleen segmentation that achieves balance between performance and model simplicity, which is of paramount importance for practical deployment.

Pros:

The paper clearly outlines the details and sources of the two datasets used in their study.

Their proposed model is a U-Net-like combining skip connections, which appears to be a good consideration for the design choice, helping to preserve vital information throughout the segmentation process.

The authors detail the computational complexity measures employed to reduce their model size and complexity.

One of the strenghts of this work is in the comparative analysis, where the authors not only describe their proposed model but also provide detailed insights into the state-of-the-art architectures against which they are compared. This provides a wide overview of the similarities and differences between the models.

The stages of pruning and quantization are carried out for both, their model and the selected state-of-the-art counterparts. This ensures a fair comparison, especially given their observation that these optimization techniques had not been previously applied to spleen segmentation in the context of the comparison models.

The references are sufficient and current.

--

Despite these contributions, there are some aspects of the methodology that prevent me for accepting the manuscript in its current form:

The paper mentions an initial hold-out data separation (70% training, 20% validation, 10% testing). However, later on, they made a statement about performing 3-fold cross-validation, but it lacks clarity regarding its integration with this initial split, leaving the reader without a clear understanding of the overall data partitioning strategy. Additionally, the rationale behind choosing 3-fold cross-validation is not explicitly said.

Authors claim that Adam was used as an optimizer for "optimal performance and faster convergence", yet this assertion is not substantiated with empirical evidence or a detailed explanation of why Adam specifically yields these benefits in their context.

While it was stated that the learning rate is not fixed, the paper lacks to describe the specific learning rate scheduling strategy thas was employed.

Authors claim that a combination of Dice Loss and Cross-Entropy Loss for optimization was employed, but the exact method or weighting of this combination is not specified.

A confusing aspect is the reporting of mIoU, which is typically reserved for multi-class segmentation tasks, as authors said. However, given that only the spleen is segmented, the justification for using mIoU is justified nor clear.

In Table 2, "Segmentation accuracy of the models," the reported IoU, mIoU, and DSC values, but there is no explicit clarification regarding the subset from which these results were obtained. Although I think they are obtained from the 10% testing subset, it is unspecified.

In Table 4, the "number of parameters" is presented as a real number rather than an integer. This value representation is not explained, although maybe an averaged value is reported, there is not further information to confirm this hypothesis.

Figure 2, the model architecture, has poor quality and legibility. The skip connections, which are a key element in their computationally efficient design, are poorly illustrated. Given their importance, a clearer visual representation of the diagram would significantly enhance the understanding of their proposed model.

Overall, the paper addresses the potential application of a simpler model, but the discussion could be significantly strengthened by analysing the clinical importance of achieving a marginally better segmentation performance, especially when compared to other models that perform similarly or occasionally better than the proposed one. In a clinical setting, slight improvements in segmentation accuracy can have substantial implications for diagnosis or treatment. The authors should at least give some insights of why these incremental gains would be particularly beneficial for real-world applications, beyond just computational efficiency, which has already been well-addressed.

**Do you want your identity to be public for this peer review?** For information about this choice, including consent withdrawal, please see our Privacy Policy

Reviewer #1: **Yes: ** José E. Valdez Rodríguez

Reviewer #2: No

---

## [Author Response · Author response to Decision Letter 1]

17 Jul 2025

Reviewer#1, Concern # 1: The authors present an optimized version of the U-Net architecture for performing semantic segmentation on spleen images, achieving improvements in both memory usage and training/inference time. However, a more detailed comparison is needed to clarify which hyperparameters were modified during the optimization process.

Author response: Thank you for your valuable comments. In the revised manuscript, first, we have clarified which hyperparameters were modified during the optimization of the traditional U-Net. These modifications are now summarized in Table 2 at the end of “The architecture of the SVNC-Net” section. The motivations behind each modification are also summarized in the table. Second, the training-related hyperparameters, such as optimizer settings, learning rate schedule, loss function, and hardware/software configurations have also been provided in Table 4 at the end of “Training and implementation” section.

Reviewer#1, Concern # 2: The authors include both numerical and visual comparisons of their results; nevertheless, several improvements are recommended for the diagrams presented. First, Figure 1 requires more detailed information: the kernel size used in each convolution and transposed convolution should be specified, as well as the spatial dimensions of the feature maps after each convolutional and deconvolutional operation. Additionally, the kernel size used for max pooling should be indicated.

Author response: Thank you for your suggestions. Figure 1 has been updated to include the kernel sizes used in all convolutional, transposed convolutional, and max-pooling operations, as well as the spatial dimensions of the feature maps after each convolutional and deconvolutional operation.

Reviewer#1, Concern # 3: It would also be highly beneficial to include a diagram comparing the original U-Net architecture with the proposed optimized version, clearly highlighting the modifications made. Furthermore, a figure illustrating the difference between standard convolution and depthwise separable convolution should be added.

Author response: Thank you for your suggestion. In the revised manuscript, the architectural differences between the original U-Net and the proposed SVNC-Net are now summarized in Table 2 at the end of the “The architecture of the SVNC-Net” section. This table provides a detailed comparison, clearly highlighting the modifications made at each architectural component. Additionally, to further address your comment, we have included Fig 2 in the same section to illustrate the difference between standard convolution and depthwise separable convolution.

Reviewer#1, Concern # 4: For Figures 4 and 5, it is recommended to label each column with a corresponding letter to make it easier to interpret the results of each experiment.

Author response: We thank you for recommendation. We have added letter labels (a–h) to each column in Figures 4 and 5 (renumbered as Figure 5 and 6 in the revised manuscript) to facilitate easier interpretation of the experimental results. The figure captions have also been updated accordingly.

Reviewer#2, Concern # 1: The paper mentions an initial hold-out data separation (70% training, 20% validation, 10% testing). However, later on, they made a statement about performing 3-fold cross-validation, but it lacks clarity regarding its integration with this initial split, leaving the reader without a clear understanding of the overall data partitioning strategy. Additionally, the rationale behind choosing 3-fold cross-validation is not explicitly said.

Author response: Thank you for your valuable comment. To clarify the data partitioning strategy, we have added a detailed explanation to the “Training and implementation” section.

Reviewer#2, Concern # 2: Authors claim that Adam was used as an optimizer for "optimal performance and faster convergence", yet this assertion is not substantiated with empirical evidence or a detailed explanation of why Adam specifically yields these benefits in their context.

Author response: Thank you for your comment. In the revised manuscript, “Training and Implementation” section has been updated to include a more detailed explanation of the motivation behind using the Adam optimizer. Specifically, we clarified its advantages in handling gradient instability and learning rate adaptation in encoder-decoder architectures, which are particularly relevant in medical image segmentation tasks involving heterogeneous input regions.

Reviewer#2, Concern # 3: While it was stated that the learning rate is not fixed, the paper lacks to describe the specific learning rate scheduling strategy thas was employed.

Author response: We appreciate the reviewer’s comment. A detailed explanation of the learning rate scheduling strategy (cosine annealing with warm restarts) that was used in the study has been added to “Training and Implementation” section in the revised manuscript.

Reviewer#2, Concern # 4: Authors claim that a combination of Dice Loss and Cross-Entropy Loss for optimization was employed, but the exact method or weighting of this combination is not specified.

Author response: We thank you for your observation. In the revised manuscript, we have provided the formulation of the total loss function in “Training and implementation” section (Eq. 5). We would like to clarify that the total loss is defined as an equally weighted combination of Dice Loss and Binary Cross-Entropy Loss. We should also note that while we used the term “Cross-Entropy Loss” in the original manuscript, we have updated it to “Binary Cross-Entropy” to more accurately reflect the binary nature of the segmentation task.

Reviewer#2, Concern # 5: A confusing aspect is the reporting of mIoU, which is typically reserved for multi-class segmentation tasks, as authors said. However, given that only the spleen is segmented, the justification for using mIoU is justified nor clear.

Author response: Thank you for your valuable comment. Although mIoU is typically used in multi-class segmentation tasks, in this study, it was computed as the average of the IoU values for the foreground (spleen) and background classes. This approach was adopted for two reasons: (1) to remain consistent with the pixel-wise classification nature of segmentation, where accurate classification of both foreground and background is essential, and (2) to evaluate the model’s overall performance and class balance, particularly in the presence of class imbalance. A brief clarification has been added to the “Mean intersection over union (mIoU)” section to explain how mIoU was calculated in the segmentation context.

Reviewer#2, Concern # 6: In Table 2, "Segmentation accuracy of the models," the reported IoU, mIoU, and DSC values, but there is no explicit clarification regarding the subset from which these results were obtained. Although I think they are obtained from the 10% testing subset, it is unspecified.

Author response: We thank you for the comment. The segmentation results in Table 2 (renumbered as Table 5 in the revised manuscript) were indeed obtained from the 10% testing subset. This has been clarified in the revised manuscript.

Reviewer#2, Concern # 7: In Table 4, the "number of parameters" is presented as a real number rather than an integer. This value representation is not explained, although maybe an averaged value is reported, there is not further information to confirm this hypothesis.

Author response: Thank you for your observation. The “number of parameters” reported in Table 4 (renumbered as Table 7 in the revised manuscript) corresponds to the total number of trainable parameters in each model. These values were expressed in millions (M) and rounded to a single decimal place for presentation clarity. We confirm that they are not averaged across runs or estimated. For instance, “60.1 M” indicates 60,103,210 parameters. This clarification has now been added as a footnote in Table 7.

Reviewer#2, Concern # 8: Figure 2, the model architecture, has poor quality and legibility. The skip connections, which are a key element in their computationally efficient design, are poorly illustrated. Given their importance, a clearer visual representation of the diagram would significantly enhance the understanding of their proposed model.

Author response: Thank you for your comments. We would like to note that the model architecture is presented in Figure 1. In the revised manuscript, this figure has been updated to clearly depict the overall architecture of the model, including the skip connections, the kernel sizes used in all convolutional, transposed convolutional, and max-pooling operations, as well as the spatial dimensions of the feature maps after each convolutional and deconvolutional layer.

Reviewer#2, Concern # 9: Overall, the paper addresses the potential application of a simpler model, but the discussion could be significantly strengthened by analysing the clinical importance of achieving a marginally better segmentation performance, especially when compared to other models that perform similarly or occasionally better than the proposed one. In a clinical setting, slight improvements in segmentation accuracy can have substantial implications for diagnosis or treatment. The authors should at least give some insights of why these incremental gains would be particularly beneficial for real-world applications, beyond just computational efficiency, which has already been well-addressed.

Author response: We appreciate for your valuable comments. Based on your suggestions, the Discussion section has been expanded by adding a new subsection “Clinical implications of segmentation accuracy” in the revised manuscript.

---

## [Decision Letter · Decision Letter 1]

1 Sep 2025

SVNC-Net: An optimized U-Net variant with 2D convolutions for lightweight 3D spleen segmentation

PONE-D-25-25158R1

Dear Dr. Dalveren,

We’re pleased to inform you that your manuscript has been judged scientifically suitable for publication and will be formally accepted for publication once it meets all outstanding technical requirements.

Kind regards,

Marco Antonio Moreno-Armendariz, Ph.D.

Academic Editor

PLOS ONE

Additional Editor Comments (optional):

Reviewer #1:

Reviewer #2:

Reviewers' comments:

Reviewer's Responses to Questions

**Comments to the Author**

Reviewer #1: All comments have been addressed

Reviewer #2: All comments have been addressed

2. Is the manuscript technically sound, and do the data support the conclusions?

Reviewer #1: Yes

Reviewer #2: Yes

3. Has the statistical analysis been performed appropriately and rigorously?

Reviewer #1: Yes

Reviewer #2: Yes

4. Have the authors made all data underlying the findings in their manuscript fully available?

Reviewer #1: Yes

Reviewer #2: Yes

5. Is the manuscript presented in an intelligible fashion and written in standard English?

Reviewer #1: Yes

Reviewer #2: Yes

Reviewer #1: The authors performed all the corrections required. With the new tables and figures, it becomes clear what the contributions of their work are.

Reviewer #2: Thank you for resubmitting your manuscript. I have reviewed the revised version and I see that the authors have addressed all of my previous comments and suggestions. The newly added "Training and implementation" and "Clinical implications of segmentation accuracy" sections provide valuable context and, in my opinion, significantly enhance the manuscript clarity and completeness.

Based on my assessment, the manuscript is now in good quality and is suitable for publication in PLOS One journal.

**Do you want your identity to be public for this peer review?** For information about this choice, including consent withdrawal, please see our Privacy Policy

Reviewer #1: No

Reviewer #2: No

---

## [Editor Report · Acceptance letter]

PONE-D-25-25158R1

PLOS ONE

Dear Dr. Dalveren,

I'm pleased to inform you that your manuscript has been deemed suitable for publication in PLOS ONE. Congratulations! Your manuscript is now being handed over to our production team.

Kind regards,

on behalf of

Professor Marco Antonio Moreno-Armendariz

Academic Editor

PLOS ONE